# In Silico Analysis and Experimental Evaluation of Ester Prodrugs of Ketoprofen for Oral Delivery: With a View to Reduce Toxicity

Kishor Mazumder [1,2,3,*], Md. Emran Hossain [1], Asma Aktar [1], Mohammad Mohiuddin [4], Kishore Kumar Sarkar [1], Biswajit Biswas [1], Md. Abdullah Aziz [2], Md. Ahsan Abid [1] and Koichi Fukase [5]

1 Department of Pharmacy, Jashore University of Science and Technology, Jashore 7408, Bangladesh; emran.du2011@gmail.com (M.E.H.); asmaaktar121039@gmail.com (A.A.); kishorekumar0811@gmail.com (K.K.S.); bb@just.edu.bd (B.B.); ahsanabid203517@gmail.com (M.A.A.)
2 School of Optometry and Vision Science, UNSW Medicine, University of New South Wales (UNSW), Sydney, NSW 2052, Australia; md_abdullah.aziz@unsw.edu.au
3 School of Biomedical Sciences, Charles Sturt University, Booroma St, Wagga Wagga, NSW 2678, Australia
4 Department of Pharmacy, Faculty of Basic Medicine and Health Sciences, University of Science and Technology Chittagong, Foy's Lake, Chittagong 4202, Bangladesh; mohiuddin.bgc90@gmail.com
5 Department of Chemistry, Graduate School of Science, Osaka University, 1-1 Machikaneyama, Toyonaka, Osaka 560-0043, Japan; koichi@chem.sci.osaka-u.ac.jp
* Correspondence: kmazumder@just.edu.bd or k.mazumder@unsw.edu.au or kmazumder@csu.edu.bd

**Abstract:** The present research aimed to synthesize ketoprofen prodrugs and to demonstrate their potentiality for oral treatment to treat chronic inflammation by reducing its hepatotoxicity and gastrointestinal irritation. Methyl 2-(3-benzoyl phenyl) propanoate, ethyl 2-(3-benzoyl phenyl) propanoate and propyl 2-(3-benzoyl phenyl) propanoate was synthesized by esterification and identified by nuclear magnetic resonance (¹HNMR) and infrared (IR) spectrometric analysis. In silico SwissADME and ProTox-II analysis stated methyl derivative as ideal candidate for oral absorption, having a >30-fold LD50 value compared to ketoprofen with no hepatotoxicity. Moreover, in vivo hepatotoxicity study demonstrates that these ester prodrugs have significantly lower effects on liver toxicity compared to pure ketoprofen. Furthermore, ex vivo intestinal permeation enhancement ratio was statistically significant (* $p < 0.05$) compared to ketoprofen. Likewise, the prodrugs were found to exhibit not only remarkable in vitro anti-proteolytic and lysosomal membrane stabilization potentials, but also significant efficiency to alleviate pain induced by inflammation, as well as central and peripheral stimulus in mice model in vivo. These outcomes recommend that ketoprofen ester prodrugs, especially methyl derivative, can be a cost-effective candidate for prolonged treatment of chronic inflammatory diseases.

**Keywords:** ketoprofen; prodrug; hepatotoxicity; gastric irritation; SwissADME; ProTox-II

## 1. Introduction

Non-steroidal anti-inflammatory drugs (NSAIDs) are considered as one of the most consumed drugs worldwide for management acute and chronic inflammation and pain associated illnesses such as osteoarthritis, ankylosing spondylitis, rheumatoid arthritis, gout, systemic lupus erythematosus and so others [1–3]. The NSAIDs group covers a large number of chemical families involving salicylates, aryl alkanoic acids, 2-arylpropionic acids or profens, fenamic acids, oxicams and sulfonamides. Rather than their variable structural and pharmacodynamics features, they exert therapeutic output by blocking cyclooxygenase enzymes (COX-I and COX-II) that regulate the prostaglandin (PG) synthesis pathway and pathogenesis of pyrexia, analgesia and inflammatory reactions mediated abnormalities [4].

Ketoprofen, chemically known as 2-(3-benzoylphenyl)-propionic acid, is a substituted 2-phenylpropionic acid (aryl carboxylic acid) derivative, or simply profen. In 1967, it

was first synthesized by Rhone Poulenc chemists (France) and introduced for therapeutic management of inflammation in France and the United Kingdom in 1973 [5,6]. At the present time, it is approved by the Food and Drug administration (FDA) as over the counter (OTC) medicine and is used to treat acute and chronic inflammatory diseases, such as rheumatoid arthritis, osteoarthritis, musculoskeletal disorders, sciatica and soft tissue injury, along with symptomatic pain management in cancer, postpartum and post-operative patients [7,8]. Along with PG inhibition, it has been reported for several activities like leukotriene synthesis inhibition, lysosomal membrane stabilization and bradykinin blocking. Remarkably a small dose of ketoprofen is capable of reducing mild to moderate analgesia to the same extent as a high dose of another profen (such as 25−50 mg ketoprofen ~400 mg of ibuprofen). Small doses of this drug are sufficient to combat non-arthritic pain (mild to moderate), whereas a large dose (about 300 mg) is required to deal with severe pain in arthritic diseases [9]. Even though it is a potent analgesic and anti-inflammatory agent, its use is limited due to severe adverse effects and toxicity, especially gastric irritation [10], renal insufficiency [11], hepatic disorder and liver toxicity [12] as well as peripheral oedema induced cardiovascular reactions [6].

Chronic inflammatory diseases (such as rheumatoid arthritis and osteoarthritis) are associated with severe inflammatory pain and require a prolonged medication therapy at high dose to ensure desired therapeutic output and regardless of major organ toxicity [13]. Instead of exhibiting prominent anti-inflammatory potential, ketoprofen is used as a second line of treatment option on account of major organ toxicities, especially gastric irritation and hepatotoxicity at higher dose (>200 mg) accompanied by lower therapeutic index (~2) and short biological half-life, $t_{1/2}$ (1.5–4 h) [14,15]. According to the Biopharmaceutics Classification System (BCS), ketoprofen belongs to BCS-II drug (low solubility and high permeability) [16]. Due to reduced solubility, a large dose is required to attain desired bioavailability via oral route, which aggravates the toxicity status. Excluding emergency condition and hospitalized coma patients, parenteral routes are not well-established owing to patient compliance, necessity of expertise healthcare professionals for administration, formulation-related higher cost (due to need of sterile condition, requirements of expertise and sophisticated machineries and packaging) and drug toxicity problems [17]. Moreover, therapeutic use of a topical dosage form of the drug is limited due to ketoprofen-induced photoallergic dermatitis [18] and poor bioavailability [19]. Esterification is a simple chemical reaction between carboxylic acid and alcohol to form ester [20]. Due to altered pKa (3.5 → 6.5) and ionization constant, ester prodrugs of weak acid, such as ketoprofen, may bypass gastric environment effectively [21,22]. The duodenum and jejunum regions of intestine metabolize the ester to form active drugs and aid in absorption [23]. Moreover, this approach is not only inexpensive, but also free from prerequisite of specialized instruments and/or conditions, expertise personnel and advanced quality control facilities.

Availability of freely accessible computational webservers (such as SwissADME and ProTox-II) permits drug development as a convenient, inexpensive, less time-consuming and more stress-free process. Theoretical prediction of drug-likeness and ADMET (absorption, distribution, metabolism, elimination and toxicity) parameters are utterly beneficial to justify the molecule as a desired drug substance before going for wet laboratory assays and in vivo studies [24,25].

Tending to attenuation of toxicities, as well as enhancement of oral bioavailability, this present investigation attempted to synthesize ketoprofen ester prodrugs and to evaluate them by means of in silico SwissADME and ProTox-II analysis, in vitro and ex vivo assays, and in vivo behavioural, as well as biochemical, studies.

## 2. Materials and Methods

### 2.1. Chemicals

Ketoprofen was purchased from Incepta Pharmaceuticals Ltd., Dhaka, Bangladesh. Sulphuric acid, dichloromethane, anhydrous sodium sulphate, ethyl acetate and n-hexane were obtained from Merck, Darmstadt, Germany. Methanol, ethanol and propanol were col-

lected from Merck, Mumbai, India. Bovine serum albumin (BSA) was procured from Sigma Aldrich, Steinheim, Germany. All the reagents and solvents used for the investigation were of analytical grade.

### 2.2. Experimental Animals

Swiss albino mice of both sex (age: 5–6 weeks and average body weight: 30–40 g) were purchased from the International Centre for Diarrheal Diseases and Research, Bangladesh (icddr,b) for in vivo behavioural experimentations. Wistar rats of male sex with 220–300 g average body weight were purchased from the animal research facility of Jahangirnagar University, Savar, Dhaka, Bangladesh for ex-vivo permeation and in vivo gastroprotective and hepatotoxicity assays. Prior to 1 week of experiments, the animals were kept in the animal house of the University of Science and Technology Chittagong (USTC) to become accustomed to the laboratory environment where standard conditions were maintained (temperature: $22 \pm 2$ °C, R.H.: $55 \pm 5$% and dark light cycle: 12 h: 12 h). They were fed icddr,b-developed standard rodent pellet food and water *ad libitum*. All the experiments were performed in accordance with the guidelines of the Institutional Animals Ethics Committee (IAEC) and study protocols (reference no. USTC/USTMEBBC/2018/12/22) were approved by the University of Science and Technology Medical Ethics, Biosafety and Biosecurity Committee (USTMEBBC) of the Basic Medical and Pharmaceutical Sciences Faculty, USTC, Bangladesh. All mice were sacrificed by euthanizing after completion of all experiments.

### 2.3. Synthesis of Ester Derivatives of Ketoprofen

Ester prodrugs of ketoprofen were synthesized by adopting the methods of Van der Bruggen et al. [20] and Ahmed et al. [26], with slight alterations. Ketoprofen (0.004 mole) and respective alcohol (0.25 mole of each methanol, ethanol, and propanol) were taken into three different flasks having a round shaped bottom, followed by connecting to a reflux condenser with the addition of concentrated sulphuric acid (Conc. $H_2SO_4$, 1 mL). The mixtures were kept in 80 °C for the esterification reaction (Scheme 1). After completing the reaction, excess alcohols were evaporated, and the extraction of synthesized crude products was carried out using dichloromethane as solvent. Finally, the extracted synthetic products were purified by column chromatography (mobile phase: 5% ethyl acetate in n-hexane).

When,
R= CH₃, Methyl 2-(3-benzoyl phenyl) propanoate (84%)
R= CH₂-CH₃, Ethyl 2-(3-benzoyl phenyl) propanoate (72%)
R= CH₂-CH₂-CH₃, Propyl 2-(3-benzoyl phenyl) propanoate (66%)

**Scheme 1.** General esterification reaction of 2-(3-benzoylphenyl)-propionic acid and alcohols to form ester prodrugs of ketoprofen.

### 2.4. Evaluation of Physicochemical Properties

The identity of the synthesized prodrugs was confirmed by thin layer chromatography (TLC), nuclear magnetic resonance (1HNMR) and infrared (IR) spectroscopy. After identification, the solubility and permeability were determined by previously-reported methods [26]. For solubility determination, excess amount of ketoprofen ester prodrugs was placed in 10 mL of phosphate buffers (pH of 7.4) at room temperature, followed by ultra-sonication (3 min). Then the solutions were allowed to stir overnight, filtered (0.45 µ) and analysed by HPLC system (column: C18, 250 × 4.6 mm; mobile phase: 42% phosphate buffer (0.1 M, pH 7.4) + 58% acetonitrile; flow rate: 0.6 mL/min, wavelength: 258 nm). The apparent partition coefficients, or Papp, of ester prodrugs was determined by saturation of

1-octanol in phosphate buffer by vigorous stirring (24 h). Then specific concentration of prodrugs was added into it, followed by attaining equilibrium by shaking for 60 min. Then the solutions were centrifuged (4000 rpm, 10 min) and concentrations were determined by the HPLC. Each compound was analysed three times.

### 2.5. In Silico Pharmacokinetic Profiling and Toxicity Analysis

2.5.1. Theoretical Prediction of Pharmacokinetic Parameters (ADME)

Physicochemical properties, drug-likeness and pharmacokinetic parameters such as ADME (absorption, distribution, metabolism, elimination) of the synthesized prodrugs were analysed using SwissADME (http://www.swissadme.ch, accessed on 20 November 2021) webserver. It is free to access and provides a robust predictive model for pharmacokinetic profiling of a drug-like compound. The 2D structures of the synthesized prodrugs were drawn by using BIOVIA Draw 2018 to create their corresponding SMILES file. Then the SMILES list of the Ketoprofen and prodrugs were entered into the SwissADME website to estimate drug-likeness parameters, bioavailability and the synthetic accessibility score, in addition to interaction with physiologically important transporters, enzymes, receptors like P-glycoprotein (P-gp) and cytochrome p-450 isoenzymes (CYPs) to induce pathogenesis [24].

2.5.2. Theoretical Prediction of Toxicity

For predicting acute as well as organ toxicity of newly synthesized prodrugs, ProTox-II (http://tox.charite.de/protox_II, accessed on 20 November 2021) web tool (free to access) was used. It incorporates molecular similarity, pharmacophores, and fragment propensities, as well as machine-learning models for the prediction of various toxicity endpoints, including acute toxicity and organ toxicities such as hepatotoxicity, cytotoxicity, carcinogenicity, mutagenicity, immunotoxicity, adverse outcomes pathways (Tox21) and toxicity targets. In this study, acute and organ toxicities were predicted by this web tool. Toxicity class and $LD_{50}$ values were also estimated [25].

### 2.6. In Vitro Dissolution Study

In vitro dissolution studies of the synthesized prodrugs were performed by using type-I dissolution apparatus (baskets apparatus) with maintaining stated conditions in USP (temperature: $37 \pm 0.5$ °C, 60 min). About 100 mg of prodrugs were placed in baskets. 900 mL of HCl buffer pH 1.2 was taken in each vessel. About 5 mL of aliquots were collected at different time intervals of 0, 5, 15, 30, 45, and 60 min and analysed by UV-visible spectrophotometer at 256 nm. A similar study was performed with phosphate buffer pH 6.8. For calculating the % of drug dissolved, the calibration curves were prepared from standard solutions of different concentrations of the respective prodrugs [27,28].

### 2.7. Ex Vivo Permeation Study

Ex vivo experimentation was conducted to assess intestinal permeability of synthesized prodrugs over rat's small intestine (duodenum) by using Ussing chamber method [29,30]. Overnight fasted rats were given access to water before sacrificing by spinal dislocation. Immediately after sacrificing, the rats were dissected and gastrointestinal tracts were excised, followed by carefully rinsing with oxygenated saline solution (*w/v*, 0.9% NaCl) by the help of a specialized blunt ended syringe. The intestinal portion (2–3 cm) was cut and mounted on Franz diffusion apparatus (HDT 1000, CPPLEY, UK) and incubated for 30 min under controlled conditions (temperature: 37 °C, aeration: 95% $O_2$/5% $CO_2$). After incubation, the synthesized prodrug (10 mg, suspended in 1 mL of phosphate buffer, pH 6.8) was transferred to the mucosal surface of a donor compartment. The acceptor compartment was filled with phosphate buffer (transport medium). A tiny magnet in the acceptor compartment was used for stirring in order to create dynamic stirring. 500 μL of aliquots were collected at predetermined time intervals for 3 h (h) from the time that the acceptor compartment and equal volume was replaced by fresh medium. The

experiment was performed three times for each prodrug. Ketoprofen was considered as the positive control. The samples were analysed by UV-Visible spectrophotometer (Double Beam Spectrophotometer U-2900/2910, Hitachi, Japan) at 258 nm wavelength and data was represented by means of apparent permeability ($P_{app}$), with a permeation enhancement ratio (R) calculated from following equations (Equations (1) and (2)):

$$\text{Apparent permeability, } P_{app} = \frac{dQ}{dt} \times \frac{1}{AC_0} \tag{1}$$

$$\text{Permeation enhancement ratio, } R = \frac{P_{app}^{(sample)}}{P_{app}^{(control)}} \tag{2}$$

where, *dQ/dt* was considered permeation flux ($\mu$g/mL), calculating from the slope of linear portion of the cumulative concentration of drug permeation versus time plot; *A* indicated the surface area of absorption barrier (1 cm$^2$); $C_0$ was the initial concentration of drugs in donor compartment ($\mu$g/mL).

*2.8. Assessment of In Vitro Anti-Inflammatory Activity*

2.8.1. Bovine Serum Albumin (BSA) Protein Denaturation Assay

BSA protein denaturation assay was carried out to assess the anti-proteolytic effect of prodrugs according to the method of Djuichou Nguemnang et al. [31], with minor alteration. BSA protein (2.7 g) was solubilized in distilled water (47.3%) with continual shaking for 1 h. Synthesized prodrug (1 mL), having concentrations of 100 and 200 $\mu$g/mL for each, were added separately to 1 mL of bovine serum albumin, followed by homogenization. The homogenates were incubated at 27 °C for 15 min and subsequently placed in a water bath (70 °C) for 10 min to denature BSA protein. Then they were allowed to cool in ambient room temperature. Pure ketoprofen and BSA homogenate were considered as standard and control respectively. Finally, activity was measured at 660 nm by UV-Visible spectrophotometer and % inhibition of BSA denaturation was calculated by following equation (Equation (3)):

$$\% \text{ Inhibition} = \frac{Absorbance^{control} - Absorbance^{sample}}{Absorbance^{sample}} \times 100 \tag{3}$$

2.8.2. Human Red Blood Cell ($^H$RBC) Membrane Stabilization Assay

Human RBC membrane stabilization assay was conducted by adopting the procedure of Gandhidasan et al. [32] and Shinde et al. [33]. Human blood was collected from healthy humans abstaining from medication therapy for a couple of weeks. After, the blood samples were stored in 4 °C for 24 h and processed for assay by mixing with Alsever solution containing dextrose (2%), sodium citrate (0.8%), citric acid (0.5%) and NaCl (0.42%) followed by centrifugation (700× *g*, 5 min). Subsequently, supernatant was discarded, and remaining cell suspension was washed using sterile saline solution (0.9% NaCl, *w/v*), followed by a couple of centrifugation and washing steps in order to obtain packed cell with colorless and transparent supernatant. Theses packed cells were suspended (40% *w/v*, 10 mM phosphate buffered saline, pH 7.4) before assay for reconstitution of cellular compartments. Prodrugs with two different concentrations, 100 $\mu$g/mL and 200 $\mu$g/mL were mixed with phosphate buffer (1 mL) and hyposaline (2 mL) and added separately to 0.5 mL of cell suspension. Hypotonicity-influenced cell membrane degeneration, as well as haemolysis, was induced by incubation (37 °C, 30 min). Afterwards, the mixtures were centrifuged (1008× *g*, 20 min), followed by collection of supernatant and analysis of hemoglobin content by UV-Visible spectrophotometer at 560 nm.

## 2.9. Assessment of In Vivo Antinociceptive and Anti-Inflammatory Activities

### 2.9.1. Acetic Acid-Induced Writhing Test

The writhing method is one of the most adopted methods to evaluate analgesic activity of synthetic or natural drugs, as well as crude medicinal substances against chemically induced peripheral pain by parenteral administration of irritants like acetic acid [34]. In this study, in keeping with the method of de Fátima Arrigoni-Blank et al. [35], the test was conducted. Mice were divided into eleven groups comprising five mice in each group. Group-I (control) and group-II (standard) received normal saline (10 mL/kg) and ketoprofen (50 mg/kg) respectively. Group-III, IV and V were treated with methyl ester of ketoprofen at the dose of 10, 50 and 100 mg/kg respectively. The rest of the six groups (group-VI to XI) were treated with 10, 50 and 100 mg/kg of ethyl ester and propyl ester of ketoprofen respectively. All the chemicals or drugs were administered in oral route. Oral doses of ketoprofen prodrugs were prepared using 1% *v/v* tween-80 as vehicle. Acetic acid was injected intraperitoneally after 30 min of drug treatment to induce peripheral pain. Subsequently, for 20 min, the muscular contractions or writhing movements were noted with a 5 min interval. The % inhibition of writhing was calculated from the following equation (Equation (4))

$$\% \text{ Inhibition } = \frac{No. \, of \, writhing^{(\, control)} - No. \, of \, writhing^{(\, sample)}}{No. \, of \, writhing^{(\, control)}} \times 100 \qquad (4)$$

### 2.9.2. Hot Plate Test

Hot plate method is a behavioral approach of estimating analgesic activity against thermal stress-induced central analgesia (acute and cutaneous pain) related to higher brain function [36]. According to the method described by Eddy et al. [37], the inhibitory effect of synthesized prodrug of ketoprofen on central analgesia was performed. Mice were grouped into eleven groups by the similar strategy of the acetic acid-induced writhing method. After 30 min of drug treatment, central pain was induced by placing the mice sequentially on a thermostatically controlled hot plate (Ugo Basile, Varese, Italy) heated with $55 \pm 1$ °C and equipped with a Plexiglas cylinder to limit their movement. The latency time designating mice behaviour on hot plate alike licking and/or shaking of hind paw or forepaw or jumping off was recorded with spontaneous removal of mice from the hot plate. The experimentation was carried out for 120 min with 30 min interval.

### 2.9.3. Formalin Induced Paw Licking Test

Formalin induced paw licking is a behavioural method of evaluating the potential of a drug or chemical substance on combating persistent pain originated from formalin sub-plantar injection [38]. In this study, this test was conducted by the method of Hunskaar et al. [39]. Mice were randomly grouped as the similar way to the writhing and hot plate methods and treated with respective drugs. Persistent pain was induced by administering sub-plantar injection of 2.5% formalin (20 μL) in one of the hind paws after one hour of drug treatment. Sterile phosphate buffered saline was injected in one of the left paw as negative control. The paw licking time was recorded as index of neurogenic pain response (early phase: 0 to 5 min) and inflammatory pain response (late phase: 15 to 30 min). The inhibitory activity was finally calculated from equation (Equation (5)):

$$\% \text{ inhibition} = \frac{Licking \, time^{(\, control)} - Licking \, time^{(\, sample)}}{Licking \, time^{(\, control)}} \times 100 \qquad (5)$$

## 2.10. Assessment of Gastroprotective Activity

Gastroprotective activity of ketoprofen prodrug was evaluated by a previously described method [40]. Wister rats were treated with ketoprofen prodrugs (100 mg/kg) by oral route for 14 days. The drugs were prepared by using 1% *v/v* tween-80 as vehicle. After 14 days, the animals were fasted overnight and sacrificed by cervical dislocation. Then the

stomach was removed, cut and examined under the dissecting microscope for observation of ulcerative lesions and scored by considering (a) normal coloured stomach: 0, (b) red colouration: 0.5, (c) spot ulcers: 1.0, (d) hemorrhagic streaks: 1.5, (e) ulcer in >50% rats: 2.0 and (f) ulcers > 80% rats: 3.0.

*2.11. Assessment of Hepatotoxicity*

The extent of hepatotoxicity produced by the ester prodrugs was evaluated by assaying liver enzymes, such as serum glutamic pyruvic transaminase (SGPT), serum glutamic oxaloacetic transaminase (SGOT), and alkaline phosphatase (ALP) as well as observing histopathology of livers of the rats treated with ester prodrugs. Wister rats of both sexes were randomly divided into five groups: group I (control): receiving 1% *v/v* tween-80 (10 mL), group II and III: receiving ketoprofen 50 and 100 mg/kg respectively, group IV and V: receiving 50 and 100 mg/kg methyl ester prodrugs of ketoprofen respectively, group VI and VII: receiving 50 and 100 mg/kg ethyl ester prodrugs of ketoprofen, respectively, and group VIII and IX: receiving 50 and 100 mg/kg propyl ester prodrugs respectively. All of the rats received respective treatments via oral route for 14 days. Oral doses of ketoprofen prodrugs were prepared using 1% *v/v* tween-80 as vehicle. After 14 days, the animals were fasted overnight and sacrificed by cervical dislocation and subjected to cardiac puncture to collect the blood. The blood was centrifuged (10 min, 3000 rpm) and the serum was analysed for assaying SGPT, SGOT, and ALP by Dimension EXM, SIEMENS automated biochemistry analyser. After sacrificing the rats, the livers were removed, washed (with ice cold saline) and tissues were portioned and fixed in 10% neutral formalin followed by dehydration, clearing and impregnation by automated tissue processor (VRX 23, SAKURA, Sakai, Japan). After that, the tissue blocks were prepared by embedding in paraffin. The blocks were cut into 5 µm sizes by microtome (LEICA RM 2125RTS, Heidelberg, Germany) and stained with hematoxylin and eosin. The stained tissue blocks were then observed and the photomicrographs of tissues were recorded by optical microscope (Olympus DP20 with BX51, 5 stations, Hamberg, Germany) [41].

*2.12. Statistical Analysis*

Statistical analysis was performed by one-way ANOVA followed by Dunnett's test ($p < 0.05$; vs. control) and post hoc Bonferroni test ($p < 0.05$; vs. ketoprofen) (SPSS software version 20; IBM Corporation, New York, NY, USA). The results were expressed as mean $\pm$ standard error of mean (SEM) and considered significant at differences of * $p < 0.05$.

## 3. Results and Discussion

### 3.1. Synthesis and Characterization of Ester Prodrugs of Ketoprofen

Esterification is considered as one of the most effective techniques of synthesizing prodrugs of weak acid offering high yield (68–89%) without complex operation, toxic solvent, sophisticated reaction vessel or conditions [42]. Ester of weak acidic medicinal substances provide an ideal approach of bypassing gastric acidic environment restricting hydrolysis, along with increasing aqueous solubility and transcellular absorption (especially in the small intestine) and rapid metabolism to produce high concentrations of the active drug in the blood to meet desired bioavailability [43]. Ketoprofen ester prodrugs were synthesized by esterification reaction (Scheme 1) using three alcohols involving reagent grade methanol, ethanol and propanol in order to obtain methyl 2-(3-benzoyl phenyl) propanoate, ethyl 2-(3-benzoyl phenyl) propanoate and propyl 2-(3-benzoyl phenyl) propanoate as finish products. After synthesis, chromatographically purified and isolated compounds were provided good yields of about 66–84% (Table S1).

Ester prodrugs of ketoprofen were analysed by [1]HNMR and IR spectroscopic techniques to characterize their chemical groups. [1]HNMR spectrum (Figure S1) synthesized by esterification of ketoprofen and methanol showed the presence of a broad singlet signal at 3.497 ppm for and multiplet signal at 7.249–7.322 ppm corresponding to methoxy proton

(–OCH$_3$) and aromatic protons (ArH), respectively, whereas the IR spectrum (Figure S2) showed characteristic bands signifying carbonyl C=O and aromatic ArC=C groups. The rest of the two prodrugs synthesized from esterification of ketoprofen with ethanol and propanol were also found to show characteristic signals and bands in $^1$HNMR and IR analysis, indicating their identity as ethyl 2-(3-benzoyl phenyl) propanoate and propyl 2-(3-benzoyl phenyl) propanoate (Table S2). All the data offered by $^1$HNMR and IR spectra were meet the characteristic functional groups and anticipated chemical structures of the three distinct ester prodrugs of ketoprofen. Solubility of ketoprofen prodrugs was increased significantly compared to ketoprofen (0.5 µg/mL). Among the three prodrugs, ethyl ester derivative was found to have the lowest solubility. Moreover, the permeability parameter, or Log P (octanol-water partition coefficient) of synthesized prodrugs was also increased compared to that of the parent drug ketoprofen (3.18) (Table S1).

### 3.2. In Silico Pharmacokinetic Profiling and Toxicity Analysis

Prior to evaluating the synthesized prodrugs in the wet laboratory, virtual screening was performed to predict whether the synthesized molecules had drugability or not. In silico web tools like SwissADME and ProTox-II are validated for estimating drug-likeness, pharmacokinetics, and toxicity of a new substance. In the drug discovery and development process, ADME analysis is considered to be one of the most challenging steps which determine the success, as well as failure, of a drug or drug-like molecule in clinical trials. It is noted that about 60% of all drugs were considered ineffective in clinical trials, simply because of the failure in prediction of drug metabolism and pharmacokinetics data [44]. ADME predictor is a computer program, designed especially for estimating pharmacokinetic profiles of drug or drug-like substances from their molecular structures [45]. Using the ADME program in SwissADME software, essential physicochemical, as well as pharmacokinetic, parameters of ketoprofen prodrugs were generated (Table 1). Molecular weight (MW) and octanol/water partition coefficient (XLOGP3) are the primary parameters of drug-likeliness filters, known as the 'rule of five', as stated by Lipinski [46]. According to the rule, at least two parameters from four basic pharmacokinetic properties (MW $\leq$ 500; XLOGP3 $\leq$ 5; no. of hydrogen bond donor $\leq$ 5; and no. of hydrogen bond acceptor $\leq$ 10.6) are obligatory to meet for being orally absorbed drug substances. MW and XLOGP3 for the synthesized prodrugs remained within the standard values of 'Rule of Five'.

**Table 1.** ADME parameters of ester prodrugs of ketoprofen representing drug likeness.

| Drugs | Drug Likeness | | | | | | | DL Score | SA Score |
|---|---|---|---|---|---|---|---|---|---|
| | MW g/mol | XLOGP3 | TPSA Å$^2$ | ESOL LogS | Fraction Csp3 | RB | BA Score | | |
| Ketoprofen | 254.28 | 3.12 | 54.37 | −3.59 | 0.12 | 4 | 0.56 | 0.57 | 2.57 |
| ME | 268.31 | 3.45 | 43.37 | −3.79 | 0.18 | 5 | 0.55 | 0.06 | 2.73 |
| EE | 282.33 | 3.81 | 43.37 | −4.02 | 0.22 | 6 | 0.55 | 0.13 | 2.94 |
| PE | 296.36 | 4.34 | 43.37 | −4.35 | 0.26 | 7 | 0.55 | 0.70 | 3.09 |

K: Ketoprofen, ME: Methyl 2-(3-benzoyl phenyl) propanoate, EE: Ethyl 2-(3-benzoyl phenyl) propanoate, PE: Propyl 2-(3-benzoyl phenyl) propanoate, MW: Molecular weight, XLOGP3: Octanol/water partition coefficient, TPSA: Topological polar surface area, ESOL LogS: Estimated aqueous solubility, Fraction Csp3: Ratio of sp3 hybridized carbons over the total carbon count, RB: Rotatable bonds, BA: Bioavailability, DL: Drug likeness, SA: Synthetic accessibility.

The 'quantitative estimate of drug-likeness', or QED, concept is currently recognized as a popular rule for quantifying drug-likeness based on eight physicochemical properties, including the four parameters described in the 'rule of five', as well [47]. MW (50–500 g/mole), XLOGP3 (2–10), topological polar surface area or TPSA (20–130 Å$^2$) and rotatable bonds or RB (0–5) are rest of the four properties. All the prodrugs met the standard values of MW (268.31 to 296.36 g/mole), XLOGP3 (3.12 to 4.34) and TPSA (43.37 Å$^2$). No. of RB in methyl 2-(3-benzoyl phenyl) propanoate was 5, which satisfies the standard value of the QED concept, whereas ethyl 2-(3-benzoyl phenyl) propanoate and propyl 2-(3-benzoyl phenyl) propanoate showed divergence. Bioavailability radar character for drug-likeness

of a molecule [48] reported an optimal range of distinct properties involving lipophilicity (XLOGP3: −0.7 to +5.0), size (MW: 150 to 500 g/mole), polarity (TPSA: 20 to 130 Å$^2$), ESOL or estimated solubility (log S: not more than 6), saturation (Fraction Csp3 or fraction of carbons in the sp3 hybridization: not less than 0.25), and flexibility (RB: not more than 9). Values of the drug likeness parameters of all the three prodrugs were found to remain within the bioavailability radar, which confirmed that all the synthesized compounds are drug substances. The bioavailability score (BA score) is a determinant of oral absorption of drug substances. Any drug molecules satisfying the 'rule of five' with a BA score of 0.55 are considered as sufficiently absorbable via oral route [49]. All of the prodrugs scored 0.55, thus demonstrating good oral bioavailability. Moreover, the prodrugs scored excellent drug-likeness (DL score) and synthetic accessibility (SA score) scores, which are considerable parameters during the drug discovery and development process.

P-glycoproteins (P-gp) are the most important ATP binding cassettes transporters or ABC transporters that mediate efflux mechanism through the gastrointestinal membrane, as well as the blood brain barrier (BBB). Any drug substance acting as P-gp substrate may cause toxicity, not only by interrupting their normal physiological functions, such as the protection of CNS, but also by controlling their expression [50–52]. Moreover, drug toxicities such as sub or supra therapeutic action, drug interaction, drug antagonism, drug accumulation, liver disorders and hepatotoxicity are directly related to the interaction of drugs with cytochrome P-450 (CYP)-mixed oxidase enzyme's major isoenzymes- CYP1A2, CYP2C19, CYP2C9, CYP2D6 and CYP3A4 [53]. Thus, it is mandatory to predict the susceptibility of new drug substances to interact with the biologically important P-gp transporter, as well as CYP enzymes. SwissADME software was used to estimate the affinity of synthesized prodrugs to be substrate of P-gp or inhibitor CYP isoenzymes; the results are represented in Table S3. Methyl 2-(3-benzoyl phenyl) propanoate showed no affinity for these proteins and isoenzymes, whereas the rest of the two prodrugs showed inhibitory effects on CYPs isoenzymes.

Acute toxicity were specific organ toxicity of synthesized prodrugs was predicted by ProTox-II webserver. Acute toxicity data (Table S2) indicated that the toxicity class of prodrugs shifted to a safer zone (toxicity class: 4) from a narrow zone (toxicity class: 2) of active ketoprofen. Predicted LD$_{50}$ values for the prodrugs were > 30 fold greater than that of active drug. Besides, organ toxicity data indicated (Table S2) that hepatotoxicity was not found in the prodrug, whereas active ketoprofen was hepatotoxic. Any other organ toxicities, such as mutagenicity, cytotoxicity, immunotoxicity and others listed in Table S2, were absent in methyl 2-(3-benzoyl phenyl) propanoate. Ethyl 2-(3-benzoyl phenyl) propanoate and propyl 2-(3-benzoyl phenyl) propanoate were found to bind with the estrogen receptor alpha (ER) whereas, other organ toxicities were absent.

### 3.3. In Vitro Dissolution Study

The dissolution study of ketoprofen ester prodrugs was conducted in both pH-simulating gastric media and intestinal media. In vitro dissolution profiles of the ester prodrugs were shown in Figure 1. The dissolution profiles indicated that release of ester prodrugs was more (~92%) in the phosphate buffer pH 6.8 than that of HCl buffer pH 1.2 (~13%). Ketoprofen was dissolved by 53.61 ± 8.9% and 81.7 ± 5.9% in HCl buffer pH 1.2 and phosphate buffer pH 6.8, respectively. In case of ester prodrugs, methyl ester prodrug dissolved the most (92.40 ± 5.9%) in phosphate buffer 6.8, whereas 7.9 ± 1.9% was dissolved in gastric media. According to USP specification, ester prodrug must dissolve less than 10% of the labelled amount after 60 min in gastric acid media and equal to and/or not less than 80% of the labelled amount after 60 min in intestinal buffer media. All the prodrugs except propyl ester prodrug satisfied the USP specifications [28]. Therefore, it can be concluded from above findings that the synthesized ester prodrugs, especially methyl ester prodrugs, comply with USP dissolution criteria for oral delivery.

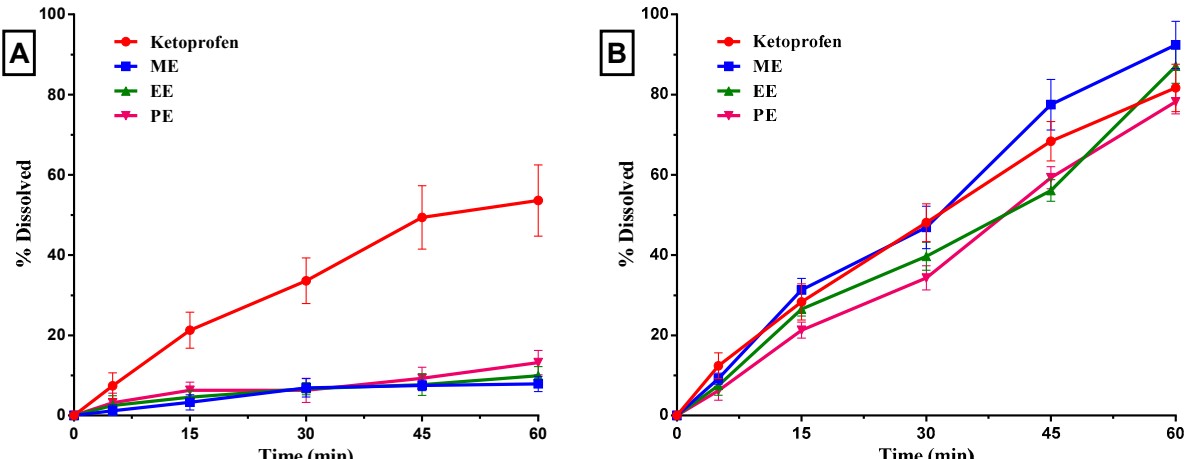

**Figure 1.** In vitro dissolution profiling of ester prodrugs of ketoprofen showing % drug dissolved in HCl buffer pH 1.2 (**A**) and phosphate buffer pH 6.8 (**B**). Data was expressed as mean ± SEM, where n = 6. ME: Methyl 2-(3-benzoyl phenyl) propanoate, EE: Ethyl 2-(3-benzoyl phenyl) propanoate, PE: Propyl 2-(3-benzoyl phenyl) propanoate.

Oral bioavailability of a drug product depends largely on its dissolution and absorption through the gastrointestinal membrane. As the synthesized prodrugs dissolve in a very small amount in the gastric pH, it will not absorbed from the stomach. Due to low solubility in HCl buffer, the gastric irritation may be reduced in comparison to pure ketoprofen. As higher solubility (> 90%) was observed in intestinal media, the esterified prodrugs can be metabolized by esterase enzymes present in the intestine and absorbed at a higher amount. Moreover, higher surface area of intestinal mucosa may accelerate the absorption many-fold, resulting in higher efficacy. Due to this higher efficacy, a minimal dose of ester prodrugs may meet the therapeutic concentration. It may eliminate the toxicity caused by higher dose, which is unavoidable in the case of a therapeutic dose of pure ketoprofen (as it is a BCS-II drug) [16].

### 3.4. Ex Vivo Permeation Study

Ussing chamber method is an innovative, flexible and inexpensive means of ex vivo technique to predict drug permeation through a biological membrane (cultured cell monolayers or excised animal tissues) [54]. Estimated apparent permeability ($P_{app}$) as well as absorption enhancement ratio (R) can predict the transport of drugs over a biological membrane, along with a mechanistic explanation [55]. In this investigation, the % permeation, $P_{app}$ and R values of the three prodrugs was observed to increase significantly compared to ketoprofen (Figure 2). $P_{app}$ of the prodrugs were $1.08 \times 10^{-7}$, $8.82 \times 10^{-8}$ and $7.84 \times 10^{-8}$ cm$^2$/s for methyl, ethyl and propyl ester derivatives consecutively, whereas ketoprofen had $6.86 \times 10^{-8}$ cm$^2$/s. A significant relative enhancement of absorption (R) was observed for methyl 2-(3-benzoyl phenyl) propanoate (1.57-fold), in contrast to pure ketoprofen. The R values for ethyl and propyl ester prodrugs were also increased compared to ketoprofen, but were relatively lower than the methyl ester prodrug. These observations suggested that ester prodrugs possessed more enhanced intestinal absorption than pure ketoprofen.

Oral absorption of a medicinal substance is affected primarily by molecular weight and partition coefficient [56]. Along with these properties, reduced molecular flexibility (denoted by number of rotatable bonds or RB) and solubility perform critical roles. The QED concept reported that the number of rotatable bond should be 0–5 for being a drug orally absorbable [47]. Aqueous solubility is also a key consideration to get the drug dissolved for reaching absorption barrier [57]. In silico drug likeness parameters showed that RB were > 5 for ethyl and propyl esters. LogS (aqueous solubility predictor) values for these two prodrugs were also decreased compared to methyl ester prodrugs. The absorption site of a drug is solely dependent on the pKa value, which is related to ionization and

drug solubility [58]. Carboxylic acid like ketoprofen has a pKa value of 3.5–4.5. This drug remains greatly unionized in stomach pH and gets absorbed, resulting severe gastric irritation. Esterification of acidic ketoprofen increases the pKa values leading to alteration of the ionization constant and solubilisation of the prodrugs at intestinal pH prominently, thus bypassing the gastric environment. The change in pKa value is the foremost aspect to increase intestinal permeation of synthesized ester prodrugs rather than pure ketoprofen.

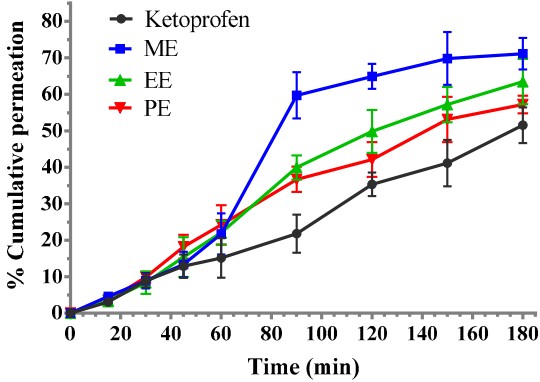

| Compound | $P_{app}$ | R |
|----------|-----------|---|
| Ketoprofen | $6.86 \times 10^{-8}$ | 1 |
| ME | $1.08 \times 10^{-7}$ | 1.57* |
| EE | $8.82 \times 10^{-8}$ | 1.29* |
| PE | $7.84 \times 10^{-8}$ | 1.14* |

**Figure 2.** % Cumulative intestinal permeation of ester prodrugs of ketoprofen assayed by ex vivo Ussing chamber method; data was expressed as mean ± SEM, where n = 3. ME: Methyl 2-(3-benzoyl phenyl) propanoate, EE: Ethyl 2-(3-benzoyl phenyl) propanoate, PE: Propyl 2-(3-benzoyl phenyl) propanoate, Papp: Apparent permeability, R: Absorption enhancement ratio. A value of * $p < 0.05$ (compared to ketoprofen) was considered significant.

### 3.5. In Vitro Anti-Inflammatory Activity

#### 3.5.1. Bovine Serum Albumin (BSA) Protein Denaturation Assay

In the BSA denaturation assay, the prodrugs were found to exhibit promising anti-proteolytic activity (Figure 3). The inhibitions were dose dependent, and the highest activity (82.61%) was observed by methyl 2-(3-benzoyl phenyl) propanoate at 200 µg/mL dose not far off ketoprofen (85.71% in 200 µg/mL dose). Ethyl 2-(3-benzoyl phenyl) propanoate and propyl 2-(3-benzoyl phenyl) propanoate showed 75.71% and 65.71% inhibition of protein denaturation at 200 µg/mL dose; whereas lower dose (100 µg/mL) of methyl 2-(3-benzoyl phenyl) propanoate showed 77.90% inhibition which was higher than the activity showed by higher dose of rest of the two prodrugs. The overall decrease of anti-proteolytic activity of the prodrugs is not statistically significant compared to standard (ketoprofen); thus, we can speculate that similar extent of desired therapeutic potential as active drugs might be obtained by the prodrugs, especially by methyl 2-(3-benzoyl phenyl) propanoate satisfying one of the basic requirements of prodrug synthesis [43]. Endogenous protein denaturation is a key pathogenic indicator of chronic inflammatory diseases like rheumatoid arthritis, osteoarthritis, serum sickness, glomerulonephritis and others [59]. Additionally, protein denaturation may lead to autoantigen generation resulting autoimmune chronic inflammatory disease, such as rheumatoid arthritis and systemic lupus erythematosus (SLE) [60]. Ketoprofen prodrugs exhibited excellent anti-proteolytic potential with lesser side effects prediction; therefore, offering a promising choice for treatment of chronic inflammatory diseases.

#### 3.5.2. Human Red Blood Cell ([H]RBC) Membrane Stabilization Assay

[H]RBC membrane stabilization assay is an established in vitro tool for predicting anti-inflammatory activity by considering the similarity of RBC membrane to lysosomal membrane [61]. In this assay, prodrugs showed significant potential to stabilize [H]RBC membrane by means of a dose dependent manner (Figure 3). Similar to BSA denaturation assay, 200 µg/mL of methyl 2-(3-benzoyl phenyl) propanoate showed most prominent stabilizing activity (81.05%) compared to ketoprofen (84.21%). Stabilizing effects of ethyl 2-(3-benzoyl phenyl) propanoate and propyl 2-(3-benzoyl phenyl) propanoate were also

considerably efficient. These findings demonstrated that prodrugs are as efficient as pure ketoprofen to stabilize $^H$RBC membrane. Generally, chronic inflammatory diseases are directly associated with extracellular activity of lysosomal enzymes, which are closed into lysosome. The degradation of lysosomal membrane leads to the release of these enzymes to induce acute or chronic inflammation [62]. In this assay, ketoprofen ester prodrugs stabilized the $^H$RBC membrane; thus, they might be effective in the stabilization of lysosomal membrane, resulting in the interruption of lysosomal enzyme release and inflammation.

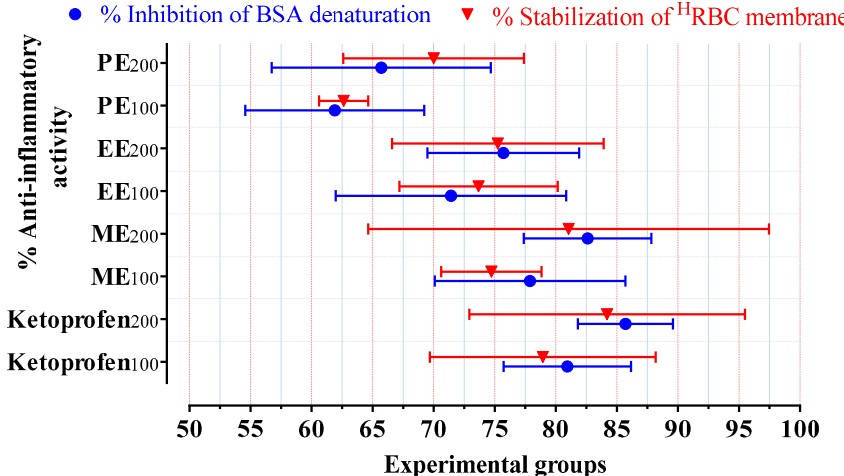

**Figure 3.** In vitro anti-inflammatory activity of ester prodrugs of ketoprofen assayed by BSA denaturation (blue) and human red blood cell haemolysis (red) techniques. Ketoprofen$_{100}$ and Ketoprofen$_{200}$: Ketoprofen 100 mg/kg and 200 mg/kg respectively, ME$_{10}$: Methyl 2-(3-benzoyl phenyl) propanoate (10 mg/kg), ME$_{50}$: Methyl 2-(3-benzoyl phenyl) propanoate (50 mg/kg), ME$_{100}$: Methyl 2-(3-benzoyl phenyl) propanoate (100 mg/kg), EE$_{10}$: Ethyl 2-(3-benzoyl phenyl) propanoate (10 mg/kg), EE$_{50}$: Ethyl 2-(3-benzoyl phenyl) propanoate Ethyl 2-(3-benzoyl phenyl) propanoate (50 mg/kg), EE$_{100}$: Ethyl 2-(3-benzoyl phenyl) propanoate (100 mg/kg), PE$_{10}$: Propyl 2-(3-benzoyl phenyl) propanoate (10 mg/kg), PE$_{50}$: Propyl 2-(3-benzoyl phenyl) propanoate (50 mg/kg) and PE$_{100}$: Propyl 2-(3-benzoyl phenyl) propanoate (100 mg/kg). Data was expressed as mean ± SEM, where n = 3.

### 3.6. In Vivo Antinociceptive Activity

3.6.1. Acetic Acid-Induced Writhing Test

Acetic acid-induced writhing is a non-selective analgesic model based on the generation of peripheral pain owing to nociceptive neurons stimulation by acetic acid induced endogenous mediators release [63]. Specified behavioural outcomes corresponding to writhing or constriction is an index of pain sensation as a result of localized inflammation regulated by the prostaglandin biosynthesis pathway [64]. No. of writhing or constrictions is reciprocal to the extent of analgesic activity [65]. In this study, oral administration of 10, 50, and 100 mg/kg of the methyl 2-(3-benzoyl phenyl) propanoate significantly decreased (*** $p < 0.001$) the no. of writhing as well as % inhibition of writhing (61.88%, 72.54% and 83.55% respectively) compared to the control, whereas standard (ketoprofen, 50 mg/kg) showed 76.83% inhibition. This activity was dose dependent and both of the ethyl 2-(3-benzoyl phenyl) propanoate and propyl 2-(3-benzoyl phenyl) propanoate followed the similar pattern of activity. Among the three prodrugs, methyl 2-(3-benzoyl phenyl) propanoate showed superior activity. The overall outcomes were presented in Figure 4. These findings confirmed that synthesized prodrugs of ketoprofen exhibit the potential to combat peripheral pain generated by the prostaglandin biosynthesis pathway, just like the reference drug ketoprofen [66].

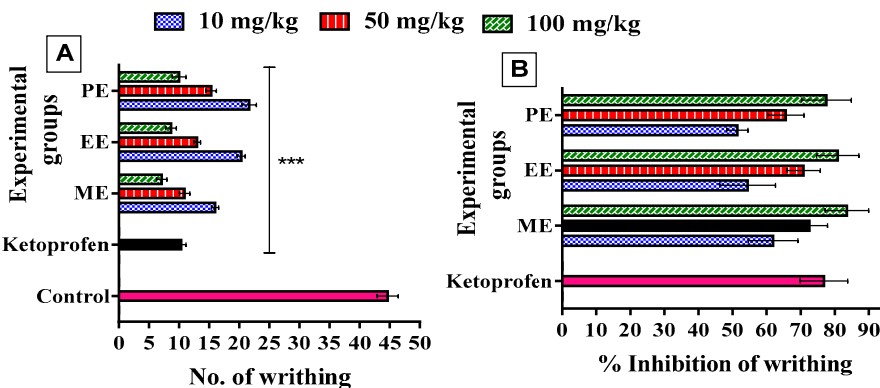

**Figure 4.** Acetic acid induced writhing test of ester prodrugs of ketoprofen, where (**A**): no. of writhing, and (**B**): % inhibition of writhing. Data are expressed as mean ± SEM, where n = 5. A value of *** $p < 0.001$ (compared to control) was considered significant.

### 3.6.2. Hot Plate Test

Hot plate is a behavioural method of evaluating the central analgesic activity by considering latency time which is directly proportional to the extent of analgesic activity. The reaction of the animal in response of thermal induced pain was considered and the time taken by the animal to react was recorded in order to estimate central analgesic activity [67]. This method is selective for the drugs that exhibit significant potential to combat inflammatory pain of central origin [68]. In Figure 5, the effects of prodrugs on hot-plate induced central analgesia in mice were shown. In this test, the reaction time observed for the 50 mg/kg and 100 mg/kg doses of prodrugs were statistically significant compared to the control. Methyl 2-(3-benzoyl phenyl) propanoate and ethyl 2-(3-benzoyl phenyl) propanoate showed a mean basal reaction time close to standard (ketoprofen, 50 mg/kg) at higher doses (≥50 mg/kg). A dose-dependent increase of latency or reaction time was observed for the prodrugs. All these findings suggested that synthesized prodrugs possessed a significant potential to reduce pain originated from the CNS.

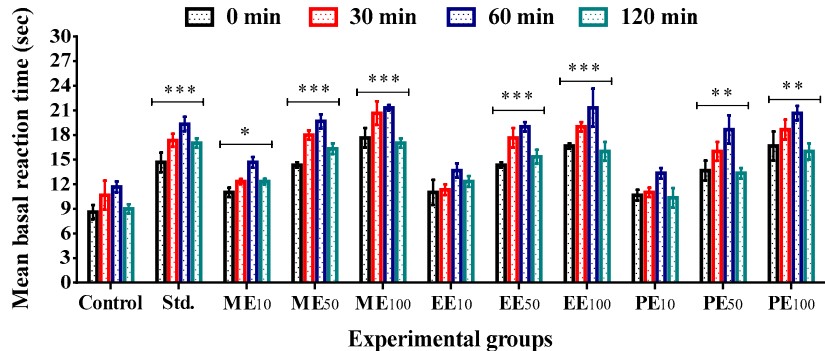

**Figure 5.** Mean basal reaction time of ester prodrugs of ketoprofen evaluated by hot plate method. Data was expressed as mean ± SEM, where n = 5. A value of *** $p < 0.001$, ** $p < 0.01$ and * $p < 0.05$ vs. control was considered significant.

### 3.6.3. Formalin-Induced Paw Licking Test

The formalin-induced paw licking is a popular method to evaluate comparison among neurogenic, central and inflammatory mechanisms of analgesia [39,69]. Formalin injection induces peripheral inflammation from central sensitization resulting release of prostaglandin, bradykinin and pro-inflammatory factors to progress neurogenic inflammation and associated pain sensation [70]. This method considers biphasic response involving early neurogenic (0–5 min, directly related to nociceptors) and late inflammatory (15–30 min, directly related to prostaglandin and pro-inflammatory factors) phase [39]. The

result of formalin induced hind paw licking test (Figure 6) indicated that the synthesized prodrugs showed dose dependent analgesic activity. The % of inhibition shown by standard (ketoprofen, 50 mg/kg) in early and late phase was 72.38% and 77.87%, respectively. In the early phase (0–5 min), methyl 2-(3-benzoyl phenyl) propanoate at the doses of 10, 50 and 100 mg/kg significantly decreased the total licking time and % inhibition (33.88%, 62.38% and 78.09% respectively) compared to the control (70.0%). A similar response was observed in the late phase (15–30 min), as well. Ethyl 2-(3-benzoyl phenyl) propanoate and propyl 2-(3-benzoyl phenyl) propanoate also showed significant decrease of licking time in both the early and late phase. The late phase response for higher doses of the prodrugs was considerably better compared to the early phase response, as with, e.g., ketoprofen. In view of all of these points, it might be concluded that synthesized ester prodrugs exhibit the potential to alleviate chronic and continuous pain originated from inflammation due to bradykinin, prostaglandin and others pro-inflammatory mediators release and leukocyte activations in inflamed tissues.

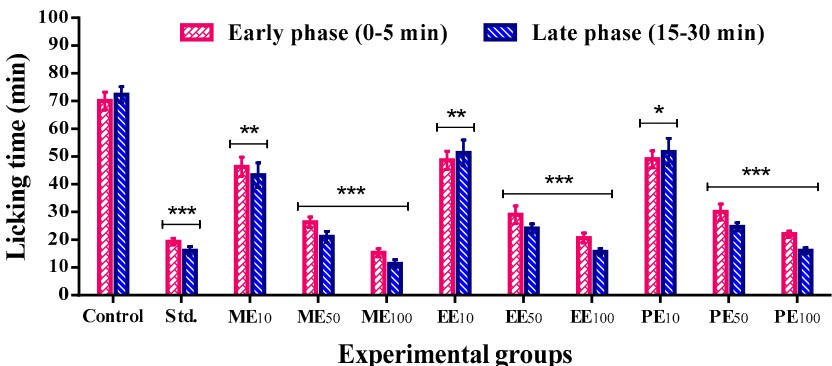

**Figure 6.** Licking time of ester prodrugs of ketoprofen assessed by using formalin induced paw licking method. Data was expressed as mean $\pm$ SEM, where n = 5. A value of *** $p < 0.001$, ** $p < 0.01$ and * $p < 0.05$ vs. control was considered significant.

### 3.7. Assessment of Gastroprotective Activity

Ketoprofen prodrugs therapy showed marked reduction of gastric erosion and ulceration in mice stomach. Under a dissecting microscope, 50% of mice treated with ketoprofen ester prodrugs were shown to have normal stomach with mild red colouration (scored as 0.5) whereas negative control mice (treated with aqueous vehicle solution) had normal coloured stomach (score: 0). Spot ulcers (1.0) were noticed in 1 mouse only. No hemorrhagic streaks were found in any mice treated with prodrugs. Apart from these, 20% spot ulceration with hemorrhagic streaks (score: 1.5) was observed in mice treated with ester prodrugs. Ketoprofen treated mice were found to have ulcerative stomach (5 mice out of 6) (score: 3). Generally, esterification approach reduces the solubility by diminishing hydrolysis of prodrugs to active form in stomach [71]. Therefore, solubility and hydrolysis of ketoprofen ester prodrugs were reduced, resulting attenuation of gastric irritation.

### 3.8. Assessment of Hepatotoxicity

Ketoprofen is an established hepatotoxic drug. It induces occurrence of perivascular infiltration and coagulation necrosis. Apart from gastric irritation, hepatotoxicity is the leading cause of diminishing its clinical use. In this investigation, the hepatotoxic potential of synthesized ester prodrugs of ketoprofen was analysed by means of examining major hepatic enzymes, as well as histopathology of hepatic sections. Figure 7 showed the observed findings of this investigation.

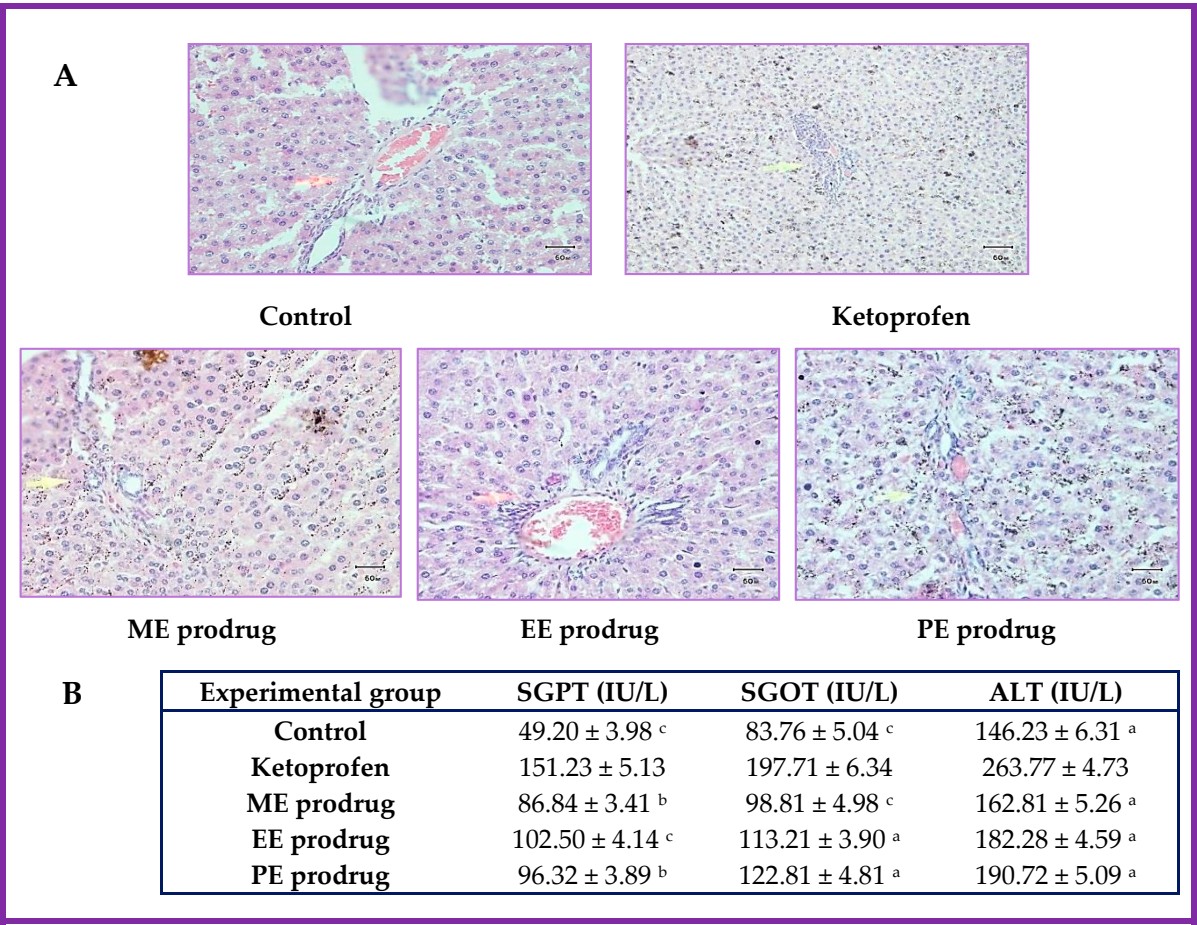

**Figure 7.** Hepatotoxicity of ester prodrugs of ketoprofen. (**A**): Histopathology of liver of experimental groups, (**B**): Liver function enzymes of experimental animals. Data was expressed as mean ± SEM, where n = 5. Values of a: $p < 0.001$, b: $p < 0.01$ and c: $p < 0.05$ vs. ketoprofen treated group were considered significant.

Three major hepatic enzymes were elevated significantly by ketoprofen treatment, whereas ester prodrugs treatments were found to affect the enzymes levels mildly (Figure 7B). As compared to ketoprofen treated group, the toxic effects of the prodrugs on liver were significantly low. Among the three prodrugs, these enzyme levels of methyl ester prodrug of ketoprofen treated rat were close to normal control. In case of histopathology (Figure 7A), the ketoprofen treated group revealed marked morphological changes, such as necrosis in the hepatic muscle fibre and infiltration with few mixed inflammatory cells. The section of ester prodrugs treated group, especially methyl ester, didn't indicate any sign of abnormalities in hepatic tissue compare to the ketoprofen group. The morphological patterns of hepatocytes in ester prodrug treated groups were similar to normal control. As the findings showed clear evidence of no hepatotoxic activity of ester prodrug treatment, it can be concluded that the synthesized ester prodrugs will be utilized therapeutically by oral route.

## 4. Conclusions

This investigation concluded that ketoprofen prodrugs synthesized by esterification could be an inexpensive and effective choice for orally active long-term therapy of chronic inflammatory conditions. In silico ADME properties, drug-likeness parameters, along with acute and organ toxicity data, signified methyl 2-(3-benzoyl phenyl) propanoate as an orally active medicinal substance with enriched oral bioavailability and reduced toxicity. Apart from these, in vivo hepatotoxicity studies evidenced that the ester prodrugs had significantly little effects on liver compared to pure ketoprofen. A significant improvement of

intestinal permeation was also revealed by ex vivo study. Excellent anti-proteolytic, as well as lysosomal membrane, stabilization activities of the prodrugs were revealed by in vitro anti-inflammatory assays. In vivo behavioural methods of analgesia established that, along with peripheral acute pain, the prodrugs were capable of alleviating inflamed tissues derived deep, as well as chronic inflammatory pain. Moreover, gastric protections were also remarkable. Hence, these findings recommended that a combination therapy of eugenol and methyl 2-(3-benzoyl phenyl) propanoate can be a suitable choice for cost-effective management of chronic inflammatory diseases offering enhanced oral bioavailability as well as reduced gastric irritation and hepatotoxicity.

**Supplementary Materials:** The following are available online at https://www.mdpi.com/article/10.3390/pr9122221/s1, Figure S1: [1]HNMR spectra of synthesized prodrugs of ketoprofen; A, B and C represented methyl 2-(3-benzoyl phenyl) propanoate, ethyl 2-(3-benzoyl phenyl) propanoate and propyl 2-(3-benzoyl phenyl) propanoate respectively, Figure S2: IR spectra of synthesized prodrugs of ketoprofen; A, B and C represented methyl 2-(3-benzoyl phenyl) propanoate, ethyl 2-(3-benzoyl phenyl) propanoate and propyl 2-(3-benzoyl phenyl) propanoate respectively, Table S1: Characterization of methyl, ethyl and propyl ester prodrugs of ketoprofen synthesized by esterification reaction, Table S2: Predicted LD50 values, toxicity class and organ toxicity of synthesized ester prodrugs of keto-profen, Table S3: Interactions of ester prodrugs with physiologically important transporters and enzymes account-able for induction of pathogenicity.

**Author Contributions:** Conceptualization, K.M. and K.F.; methodology, K.M., M.E.H. and A.A.; software, A.A. and M.A.A. (Md. Ahsan Abid); validation, K.M. and A.A.; formal analysis, K.M., M.M. and A.A.; investigation, M.M. and M.A.A. (Md. Ahsan Abid); resources, K.M., M.E.H. and A.A.; data curation, M.M., K.M. and M.A.A. (Md. Ahsan Abid); writing—original draft preparation, A.A., M.E.H. and K.M.; writing—review and editing, K.M., K.F., K.K.S., M.A.A. (Md. Abdullah Aziz) and B.B.; visualization, A.A., B.B., K.K.S. and M.A.A. (Md. Abdullah Aziz); supervision, K.M.; project administration, K.M.; funding acquisition, K.M. and K.F. All authors have read and agreed to the published version of the manuscript.

**Funding:** This research received no external funding.

**Institutional Review Board Statement:** The study was conducted according to the guidelines of the Institutional Animals Ethics Committee (IAEC) and study protocols (reference no. USTC/USTMEBBC/2018/12/22) were approved by the University of Science and Technology Medical Ethics, Biosafety and Biosecurity Committee (USTMEBBC) of the Basic Medical and Pharmaceutical Sciences Faculty, USTC, Bangladesh.

**Informed Consent Statement:** Not applicable.

**Data Availability Statement:** Not applicable.

**Conflicts of Interest:** The authors declare no conflict of interest.

## Abbreviations

ME: Methyl 2-(3-benzoyl phenyl) propanoate; EE: Ethyl 2-(3-benzoyl phenyl) propanoate; PE: Propyl 2-(3-benzoyl phenyl) propanoate; NMR: Nuclear Magnetic Resonance; IR: Infrared Spectroscopy; NSAIDs: Non-steroidal anti-inflammatory drugs; PG: Prostaglandin; COX: Cyclooxygenase; BCS: Biopharmaceutics Classification System; ADMET: Absorption, Distribution, Metabolism, Elimination and Toxicity; BSA: Bovine serum albumin; TLC: Thin Layer Chromatography; P-gp: P-glycoprotein; CYPs: Cytochrome p-450 isoenzymes; Papp: Apparent permeability; R: Absorption enhancement ratio; [H]RBC: Human Red Blood Cell; SGPT: Serum glutamic pyruvic transaminase; SGOT: Serum glutamic oxaloacetic transaminase; ALP: Alkaline phosphatase; QED: Quantitative estimate of drug-likeness; MW: Molecular weight; XLOGP3: Octanol/water partition coefficient; TPSA: Topological polar surface area; ESOL LogS: Estimated aqueous solubility; Fraction Csp3: Ratio of sp3 hybridized carbons over the total carbon count; RB: Rotatable bonds; BA: Bioavailability; DL: Drug likeness; SA: Synthetic accessibility; BBB: Blood brain barrier; and SLE: Systemic lupus erythematosus.

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
