# Peer review of "In Silico Analysis and Experimental Evaluation of Ester Prodrugs of Ketoprofen for Oral Delivery: With a View to Reduce Toxicity"

_processes, doi:10.3390/pr9122221_

Round 1

Reviewer 1 Report

This article by Dr. Mazumder and co-workers “In silico analysis and experimental evaluation of ester prodrugs  of ketoprofen for oral delivery: with a view to reduce toxicity “ presents the ketoprofen prodrugs synthesis and its potentiality as oral treatment to treat chronic inflammation reducing its hepatotoxicity and gastrointestinal irritation.

In the supporting material a list of chemical compounds and its chemical structure after the NMR spectra should be very supportive. Besides NMR and FTIR quality spectra is recommended.

Overall, the work presented is very comprehensive, having performed a complete analysis.

The manuscript structure is appropriate but some re-writing is required to improve readability.

I recommend publication of the work after some changes and English editing is performed.

Author Response

Thank you so much for your kind review and comments.  We tried our best to follow your these . Please find our response as below:

In the supporting material a list of chemical compounds and its chemical structure after the NMR spectra should be very supportive. Besides NMR and FTIR quality spectra is recommended.

Response:  We have incorporated structures of each compound in table S1 of the supplementary file. We tried to improve the quality of the spectrum.

The manuscript structure is appropriate but some re-writing is required to improve readability.

Response: To improve readability, some lines were re-written. careful English revision was made one the competent authors.

The NMR and FTIR spectra have been included in the supporting materials.

Reviewer 2 Report

        The article - "In silico analysis and experimental evaluation of ester prodrugs 2 of ketoprofen for oral delivery: with a view to reduce toxicity" covers an important aspect of the side effects of drugs, including the commonly used NSAIDs (ketoprofen). In order to eliminate the adverse effects, the authors proposed a new esterified form of ketoprofen - the drug with better properties and a lower impact on the digestive system.The authors present the subject broadly through the use of in vivo, in vitro and in silico study models.The article is very interesting, but I have some comments: 
MINOR mistakes:
1.  There are many abbreviations used in the article, it would be worth introducing their list.
2.  No consistency in the markings in the figures once ketoprofen is called Sts. and once a whole name. Similarly, the ester derivatives were used once in full names and sometimes abbreviations. This could be made uniform. This could be made uniform.
3.  I do not understand the statistics (Figure 4) against the pure ketoprofen drug since only methyl products show the statistical significance and the rest of the ester products do not?  
MAJOR mistakes:
1.  No information about statistical tests used in the study, please describe the statistical analysis in the methodology.
2.  No discussion, please add this part or compare the results with other studies in the result part - create a chapter results and discussion.
3.  Chapter 3.3 describes the solubility of newly formed drugs in HCl and phosphate buffer. The low solubility of the derivatives in HCl has been demonstrated. However, its effect on drug efficacy is not described.
4.  Figure 7A - no description of the three photos. Only control and ketoprofen are described. Please correct it.
5.  Figures 4, 5, and 6 - ketoprofen used only in one concentration (additionally, no description in the graphic), unlike ester products. It's hard to compare a concentration of 50 mg/kg to a concentration of 100 mg/kg. Why is “ketoprofen 100” not on the figure?

Author Response

Thank you so much for your kind review and comments.  We tried our best to follow these . Please find our response as below:

MINOR mistakes:

  1. There are many abbreviations used in the article, it would be worth introducing their list.

Response: List of abbreviations was introduced after conclusion section.

  1. No consistency in the markings in the figures once ketoprofen is called Std. and once a whole name. Similarly, the ester derivatives were used once in full names and sometimes abbreviations. This could be made uniform.

Response: Corrections were made as per instructions.

  1. I do not understand the statistics (Figure 4) against the pure ketoprofen drug since only methyl products show the statistical significance and the rest of the ester products do not?  

Response: The comparison was made against control. The activities of the synthesized prodrugs were better than ketoprofen but not statistical significant.

MAJOR mistakes:

  1. No information about statistical tests used in the study, please describe the statistical analysis in the methodology.

Response: Statistical analysis was described in the methodology section.

  1. No discussion, please add this part or compare the results with other studies in the result part - create a chapter results and discussion.

Response: Result and discussion part was created as per instruction.

  1. Chapter 3.3 describes the solubility of newly formed drugs in HCl and phosphate buffer. The low solubility of the derivatives in HCl has been demonstrated. However, its effect on drug efficacy is not described.

Response: The effect of low solubility of the prodrugs in HCl buffer on drug efficacy was described as per suggestion.

  1. Figure 7A - no description of the three photos. Only control and ketoprofen are described. Please correct it.

Response: Correction was made as per instruction.

  1. Figures 4, 5, and 6 - ketoprofen used only in one concentration (additionally, no description in the graphic), unlike ester products. It's hard to compare a concentration of 50 mg/kg to a concentration of 100 mg/kg. Why is “ketoprofen 100” not on the figure?

Response: As effective oral dose of ketoprofen, 50 mg/kg is established. The synthesized prodrugs were given at a dose of 50 mg/kg to compare the effects of the drugs with standard drug ketoprofen. The additional two doses (one higher than 50 mg//kg and another one lower than 50 mg/kg) of synthesized prodrugs was introduced. These two additional doses (10 mg/kg and 100 mg/kg) were not used to compare with ketoprofen rather it was introduced to compare the effectiveness of different doses of the ester pro-drugs and to establish a dose dependent activities of the synthesized drugs.

Round 2

Reviewer 2 Report

The authors' corrections are acceptable. I believe that the publication may be published in a journal.